# Any2Policy: Learning Visuomotor Policy with Any-Modality

**Yichen Zhu, Zhicai Ou, Feifei Feng, Jian Tang**[*]
Midea Group

## Abstract

Humans can communicate and observe media with different modalities, such as texts, sounds, and images. For robots to be more generalizable embodied agents, they should be capable of following instructions and perceiving the world with adaptation to diverse modalities. Current robotic learning methodologies often focus on single-modal task specification and observation, thereby limiting their ability to process rich multi-modal information. Addressing this limitation, we present an end-to-end general-purpose multi-modal system named Any-to-Policy Embodied Agents. This system empowers robots to handle tasks using various modalities, whether in combinations like text-image, audio-image, text-point cloud, or in isolation. Our innovative approach involves training a versatile modality network that adapts to various inputs and connects with policy networks for effective control. Because of the lack of existing multi-modal robotics datasets for evaluation, we assembled a comprehensive real-world dataset encompassing 30 robotic tasks. Each task in this dataset is richly annotated across multiple modalities, providing a robust foundation for assessment. We conducted extensive validation of our proposed unified modality embodied agent using several simulation benchmarks, including Franka Kitchen and Maniskill2, as well as in our real-world settings. Our experiments showcase the promising capability of building embodied agents that can adapt to diverse multi-modal in a unified framework.

## 1 Introduction

What is the ultimate form of robots? It should possess the ability to both listen and read, learn from demonstrations, and perceive the three-dimensional world with an understanding of time. While recent advancements in robot learning have explored various modalities for task specification and environmental perception – such as text, speech, images, videos, and point clouds – most prior research has approached these modalities as distinct challenges.

An expanding field of research within artificial intelligence, spanning various disciplines, indicates that simultaneous learning across different modalities, such as vision-language [1, 2, 3, 4, 5], video-text [6, 7, 8, 9, 10, 11], and vision-touch [12, 13, 14, 15], leads to the development of more comprehensive and effective representations. The enriched representations derived from this approach lead to a more profound comprehension of each modality on a standalone basis. This multi-faceted concept finds backing in a number of cognitive science and psychology studies [16, 17], which propose that human learning is significantly enhanced when it integrates multiple cues, compared to relying on a single modality in isolation. By adopting this multi-modal approach, which leverages the collective strength of various senses or inputs, we emulate the natural human method of information acquisition and processing, indicating a path toward a more integrative methodology in robotics.

---

[*]Corresponding Author

38th Conference on Neural Information Processing Systems (NeurIPS 2024).

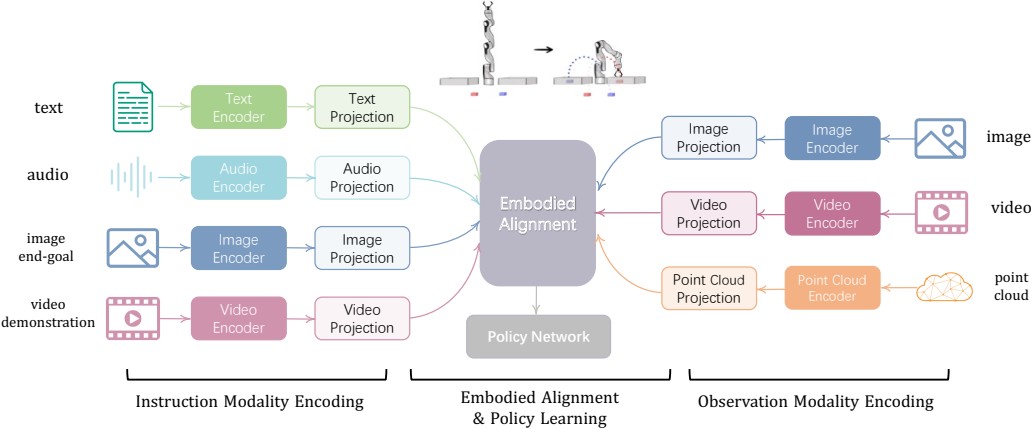

Figure 1: The overall framework of Any2Policy. The Any2Policy framework is structured to handle multi-modal inputs, accommodating them either separately or in tandem at the levels of instruction and observation. We design embodied alignment modules, which are engineered to synchronize features between different modalities, as well as between instructions and observations, ensuring a seamless and effective integration of diverse input types.

Building on the proven success of multi-modal learning in various disciplines, our goal is to develop a multi-modal embodied agent capable of processing a wide array of instruction and observation modalities. Our method, referred to as Any-to-Policy Robot Model (abbreviated as Any2Policy), is illustrated in Figure 1. The Any2Policy architecture is structured in two primary components. Initially, we harness multi-modal encoders to extract relevant features from the input. These features are then transformed into standardized token representations, ensuring uniformity at the neural network level via projection layers. In the second component, we introduce embodied alignment, which processes multi-modal signals, enriched with specific instructions. These signals are routed to encoders following projection, ultimately leading to the generation of actions for manipulation.

In support of this project, we are releasing a substantial real-world dataset consisting of 30 tasks, where each task includes 30 trajectories, all annotated with multi-modal instructions and observations, mirroring the setup used in our experiments. The purpose of this dataset is to foster and encourage future research in the area of multi-modal embodied agents.

We have demonstrated the efficacy of our Any2Policy approach through rigorous testing against this benchmark. The results clearly indicate that Any2Policy is highly adept at interpreting and responding to a variety of modalities. We further conduct experiments on two simulated robotics benchmarks to reinforce the strong generalizability of our approach compared to existing methods. Our project is at any2policy.github.io/.

In summary, our contributions are the follows:

- We introduce any-to-policy models that enable a unified embodied agent to process various combinations of modalities, effectively facilitating instruction and perception of the world.
- We present novel embodied alignment learning techniques designed to seamlessly align instructions and observations, enhancing both the effectiveness and efficiency of policy learning.
- We offer a multi-modal dataset tailored for robotics, encompassing 30 distinct tasks. This dataset covers a wide spectrum of modalities in both instruction and observation.

## 2   Related Work

**Multi-Modal Learning.** Recent large vision language models (VLMs) use pretrained image encoders as part of the larger model; some pretrain it with supervised classification, some use the pretrained CLIP encoders [18], and some with custom multi-modal pretraining. Inspired by GPT-4, which has been released and demonstrates many advanced multi-modal abilities, recent models present more

lightweight versions of VLM that align the image inputs with a large language model using proper instructional tuning [19, 20, 21, 22]. These vision-language models also showcase many advanced multi-modal capabilities after the alignment. Another line of research [23, 24, 25, 26, 27, 28] seeks to demonstrate the transformer's capability to deal with multiple modalities in a single neural network and then perform image generation, video-question answering, and multi-turn dialogue, graph learning [29, 30, 31] according to the input modalities. Different from these prior works, we focus on robotics. Our design multi-modal input includes both instruction and observation, which potentially makes the robot more generalizable to diverse scenarios.

**Embodied agent with Diverse Modalities.** The study of embodied AI has been extensively explored through instruction and observation methods. A series of studies concentrate on learning using varied task specifications. This includes language-conditioned policy learning [32, 33, 34, 35, 36, 37, 32, 38, 39, 40], instruction following agents [41, 42] with 3D observation [43, 44, 45, 46, 47], visual goal-reaching [48, 49, 50, 51], video demonstrations [52, 53, 54, 55, 56, 34, 57, 58], or combination with multiple modalities [59, 60, 61]. Notably, most systems are specialized for certain task specifications, with few like VIMA and MUTEX engaging with multiple modalities, albeit focusing on a single observation type.

Representation learning is critical in high-dimensional control settings, particularly when managing visual observation spaces. Recent advancements in this field have explored different modalities in feature learning for robotics, especially for images with the unsupervised pretrained visual representation [62, 63, 64, 65, 66, 67, 68, 69], or using visual encoders from pretrained vision-language models [70, 71, 72, 73, 74, 75, 76, 77] for robotics [78, 79, 80, 33, 81, 82, 59, 83, 84, 61]. These approaches often neglect instruction or rely solely on text-based instruction, lacking exploration of diverse instructional modalities.

Our proposed Any2Policy framework differs from previous work by focusing on both instruction and observation levels and utilizing various modalities. This approach aims to enhance learning and inference for embodied agents, offering a more holistic and integrated perspective compared to earlier, more modality-specific research.

**Robot Manipulation and Datasets**. A diverse array of robot manipulation tasks necessitates various skills and task specification formats, including instruction following [32], one-shot imitation [85], rearrangement [86], reasoning [87, 88, 89], and planning [90, 91]. Most existing datasets predominantly feature 2D images for observations and, in some cases, include language instructions to prompt specific robotic actions, as seen in datasets like Franka-Kitchen [92] and CALVIN [93]. VIMA-Bench [59] introduces a dataset that combines language instructions with images in multimodal prompts. Meanwhile, Maniskill [94] provides 3D visual information, enabling models to interpret point clouds as observations. Our dataset, RoboAny, stands out as the first to support a comprehensive range of modalities in robotics. It encompasses both instructions and observations across images, videos, audio, language, and point clouds, offering unparalleled diversity in robotic task specification and learning.

## 3 Methodology

Our objective is to develop a comprehensive policy capable of executing a variety of tasks, utilizing a dataset of demonstrations that are annotated with multi-modal instructions and observational data from various sources such as images, videos, point clouds, text, and speech. We acknowledge that the modality of task specifications and observations may differ depending on the specific scenario, and our approach is tailored to adapt flexibly to these varying modalities.

We start with a formal description of our settings and provide mathematical annotations. For each task $T \in \{T_1, T_2, \cdots, T_n\}$, we assume that the embodied agents learn from demonstrations obtained through teleoperation, $D_i = \{d_i, \cdots, d_m\}$, where $m$ is the number of demonstrations for task $T_i$. Each demonstration $d_i^j$ presents the form of a sequence of observations and expert actions, $d_i^j = [(s, o, a)_i^j]$, where $a \in A$ denotes the actions, $s \in S$ represent instruction, and $o \in O$ is the observation.

The instruction of task $T_i$ is specified with four modalities: text instructions $s \in \{L_i^1, L_i^2, \cdots, L_i^k\}$, audio instruction $s \in \{A_i^1, A_i^2, \cdots, A_i^k\}$, image goal instruction $s \in \{M_i^1, M_i^2, \cdots, M_i^k\}$ and video demonstration $s \in \{V_i^1, V_i^2, \cdots, V_i^k\}$. Similarly, the observation of task $T_i$, is detailed by

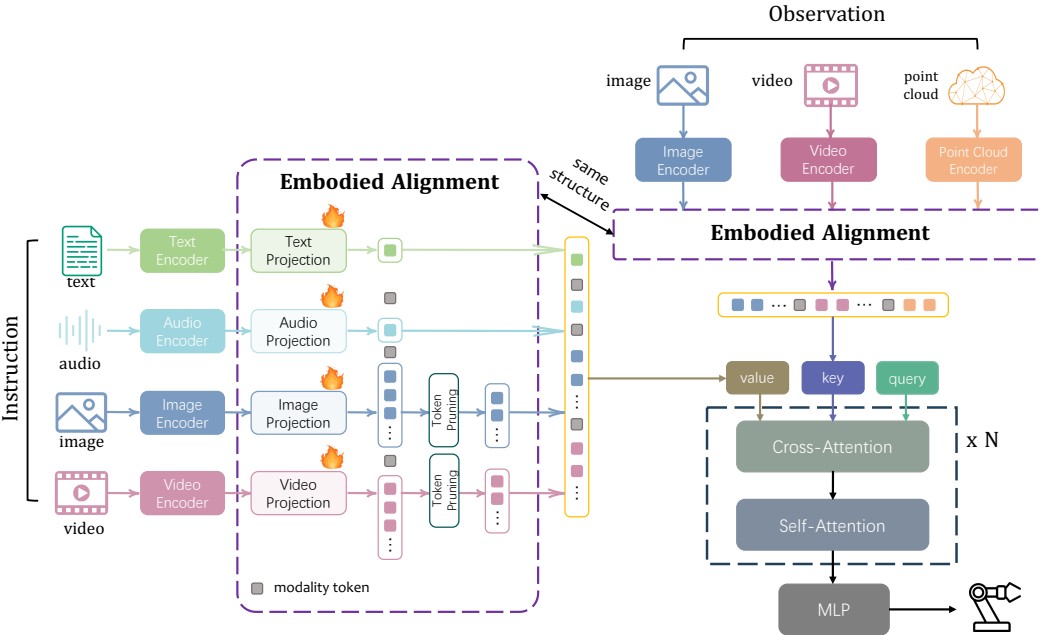

Figure 2: The architecture of embodied alignment and policy network.

$m$ alternative perceiver, each with three modalities: image $o \in \{m_i^1, m_i^2, \cdots, m_i^k\}$, point cloud $o \in \{p_i^1, p_i^2, \cdots, p_i^k\}$, and video $o \in \{v_i^1, v_i^2, \cdots, v_i^k\}$. Capital letters are used to denote instructions, while lowercase letters represent observations.

Our objective is to develop a multi-task policy, denoted as $\pi(a|s, o)$, which generates continuous actions $a$ in response to the current observation $o$ and given instruction $s$. The policy aims to proficiently execute $n$ tasks included in the training dataset $D$ under new initial conditions, such as varying positions of objects. Moreover, a critical aspect of this policy is its capacity to generalize effectively to environments, entities, and tasks that have not been previously encountered or represented in the training data.

## 3.1 Overall Architecture

A pivotal characteristic of an Any2Policy agent is its capacity to leverage multi-modal information throughout its training phase, exhibiting the ability to operate competently when restricted to a single modality. It adapts its comprehension and actions to suit the available sensory input. This versatility brings complexities in crafting multi-modal models, requiring careful consideration in their design to ensure seamless adaptability and robust performance. To address these, we introduce two modules: 1) Multimodal encoders to promote cross-modal interactions of all modalities into a shared latent space for instruction and observation, individually, and 2) Embodied alignment to take in the instruction token and merged with observation tokens via cross-attention. We provide an overview in Figure 2.

**Multimodal Encoder.** There are many choices for the multi-modal encoders, for instance, CLIP [18] for language and vision transformer for images. Specifically, we denote the encoder as a mapping function $P(\cdot)$, and for each modality, we need a particular model to extract useful modality-agnostic representation to prepare for policy learning. To make our framework more consistent, we would like to share the encoder if the instruction and observation have the same modality. Then, we would have five modality encoders for text, audio, image, video, and point cloud, respectively.

Using different backbone models for each modality can be cumbersome and hard to maintain. Thus, we leverage existing well-established models, ImageBind [25], a unified high-performance encoder across five modalities, to encode inputs of various modalities. With the help of ImageBind, we are spared from managing many numbers of heterogeneous modal encoders. Notably, the multi-modal encoders extract semantically meaningful representations from the input modality, which aims to provide essential knowledge to the policy learning networks for accurate manipulation.

**Embodied Alignment.** We leverage a projection layer to map different modalities' feature representations from $P(x)$ into unified representations. In particular, we want to avoid heavy computational costs during training, so that we freeze all the layers in ImageBind, and only keep the projection layers learnable. This operation is kept the same for both the instruction segment and the observation segment.

As a result, the projection layers ensure that the tensor of different modalities are in the same form. Nevertheless, the process is fraught with challenges, 1) aligning distinct modalities within a segment, particularly when diverse types such as images and point clouds coexist, is difficult. Simultaneously, it's vital to control computational costs, which can escalate with the addition of modalities, and 2) the integration of instructions with observations is complex. This complexity arises partly because instructions and observations don't always correspond one-to-one; often, a single instruction relates to multiple observation frames, and for most time steps, observations exist without accompanying instructions. Additionally, the modality gap between instruction and observation can be an obstacle in creating effective feature representations for manipulation tasks.

To address the first challenge, we employ a transformer architecture. Each modality undergoes tokenization, with the token count varying between modalities. For example, images and videos typically require more tokens to encapsulate visual information. We use 81 tokens for images, $81 \times t$ for videos (where $t$ represents the number of frames), and 256 tokens for point clouds. For text and audio, a single token is generally sufficient due to their shorter length. It's important to note that the computational demands of self-attention layers increase quadratically with the number of tokens, leading to potentially high inference costs for visual modalities. Therefore, it is crucial to use model compression methods [95, 96, 97, 98, 99, 100, 101, 102, 103, 104] to reduce the computational cost.

We follow RT-1 [35] to utilize TokenLearner [105] for pruning redundant visual tokens. We apply TokenLearner individually to each visual modality, such as images, videos, and point clouds. After token pruning, the token count for images and videos is reduced to 8, while for point clouds, it's brought down to 16. These tokens are then amalgamated, using a modality token to distinguish between different modalities. We insert a modality token between every two modalities. We also use absolute position embedding to maintain the token order. Consequently, the maximum token count, when integrating all three visual modalities, is significantly reduced to as few as 34 (8+8+16+2), substantially enhancing computational efficiency. For the transformer model, we use one Transformer block with self-attention and feed-forward networks.

The second major challenge is how to align instruction with observation. We seek the help of cross-attention. Specifically, when we are presented with an observation sequence, $P_o$, and an instruction sequence, $P_s$, we project $P_o$ into the key (K) and $P_s$ into the value (V). We create a set number of learnable query (Q) embedding as input for the cross-attention. This learnable query embedding aims to adapt different modalities for efficient fusion between instruction and observation. This setup enables us to align the two tensors through a cross-attention mechanism, mathematically represented as:

$$h_i = Softmax(\frac{Q_i K_i^T}{\sqrt{d_h}} V_i) \tag{1}$$

In this equation, $d_h$ denotes the dimensionality, and the construction of the query, key, and value components aligns with the standard framework utilized in BERT [106]. The elegance of the method lies in its ability to effectively incorporate relevant representations from the instruction into the observation, thus ensuring a more coherent and informative interaction between the two. In our experiments, we use 16 queries, where the dimension of queries corresponds to the cross-attention layers.

**Policy Networks.** Designing a multi-task policy comes with its share of challenges, and one critical aspect involves choosing an appropriate conditioning mechanism. In our framework, the policy networks are conditioned based on the observation sequence and occasionally an instruction sequence. As aforementioned, this conditioning is achieved through a series of cross-attention layers, which establish connections between the policy network and the input modalities. Each Transformer block comprises self-attention, cross-attention, and a feed-forward network. Additionally, we incorporate residual connections that link the higher layers with the input rollout trajectory sequence. We append one history action token to the model, similar to VIMA approach [59]. It is important to note that the instruction aligns with the observation at every time step. To optimize efficiency, we store the instruction's representation to avoid repetitive computations. The decoder is constructed using $L$

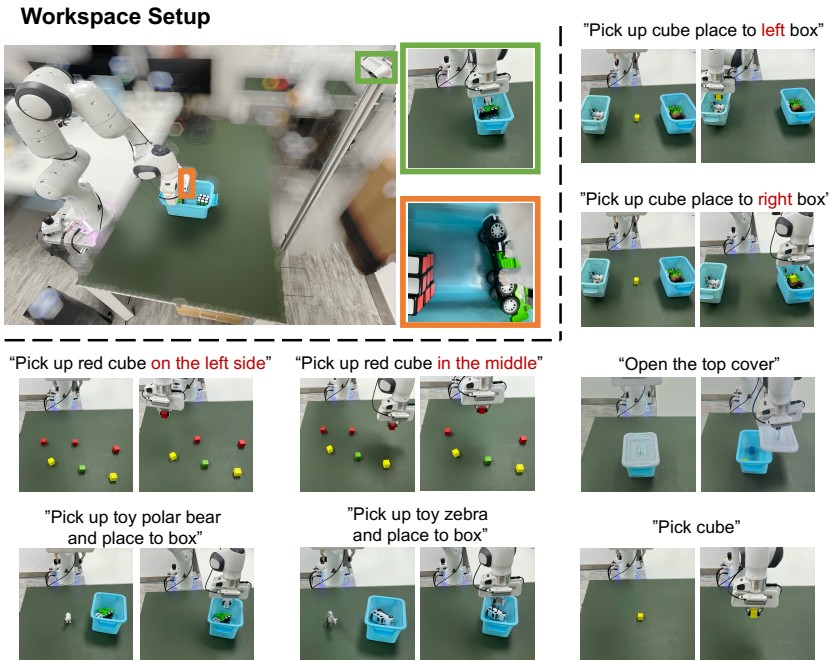

Figure 3: This is the setup of our Franka real robot. We have compiled examples of several tasks. To facilitate better understanding for our readers, we provide only the language-based versions of these task descriptions.

Transformer blocks. Finally, we append a Multi-layer Perceptron (MLP) layer to generate continuous actions. We use behavior cloning mean squared loss as the optimization objective.

## 3.2 Dataset Construction

In our pursuit to develop a robot model that allows input with any modality, we have established a new dataset featuring a range of tasks, each accompanied by multiple instructions and observations tailored to real-world scenarios. We developed two workstations, one of which is depicted in Figure 3. This dataset comprises $n = 30$ distinct tasks. These tasks vary from straightforward pick-and-place actions, such as "pick up the yellow cube," and "place the toy bear in the blue box to the right," to more complex contact-rich tasks like "open the drawer" as well as tasks demanding to reason, for instance, "sort the cube with the same color." Although all tasks are executed within the same environment, they involve a variety of objects selected from a set of 70. Each task is exemplified through $m = 30$ human-collected trajectories. Further, every task is annotated with $k = 5$ distinct instructions. To generate these diverse instructions, we utilized GPT-4, creating prompts to formulate alternative descriptions. We have carefully filtered these descriptions to ensure they accurately represent the specific task-relevant objects, avoiding any potential confusion due to synonyms. Additionally, we have incorporated speech signals from various characters of the Amazon Polly service to enrich our dataset with diverse speech descriptions. In terms of observation, we recorded the video during our data collection process. The depth information captured in these recordings was then converted into point cloud data, enabling detailed analysis and application of the collected data. All data are collected via teleoperation. For long-horizon tasks, we operated the robot to move back to the initial state and then start the next move. In the appendix, we provide detailed descriptions of our hardware for two workstations and more qualitative examples of our collected data.

## 4 Experiments

This section aims to demonstrate the generalizability of our proposed Any2Policy agent. We evaluate our method in two setups. Initially, we conduct experiments in a real-world setting using our own collected dataset, which comprises various modalities. This is aimed at demonstrating the efficacy

Table 1: Comparison with modality-specific models in real-world experiments.

| Instruction → Observation | Text → Image | Text → Video | Text → Point Cloud | Audio → Image | Audio → Video | Audio → Point Cloud |
|---|---|---|---|---|---|---|
| **Any2Policy** | **51** | **57** | **62** | **49** | **55** | **57** |
| Modality-Specific | 39 | 42 | 47 | 26 | 48 | 50 |

| Instruction → Observation | Image → Image | Image → Video | Image → Point Cloud | Video → Image | Video → Video | Video → Point Cloud |
|---|---|---|---|---|---|---|
| **Any2Policy** | **56** | **63** | 38 | **46** | **57** | **36** |
| Modality-Specific | 45 | 47 | **45** | 39 | 46 | 28 |

Table 2: Comparison with Any2Policy framework without embodied alignment in real-world experiments

| Instruction → Observation | Text → Image | Text → Video | Text → Point Cloud | Audio → Image | Audio → Video | Audio → Point Cloud |
|---|---|---|---|---|---|---|
| **Any2Policy** | **51** | **57** | **62** | **49** | **55** | **57** |
| - Embodied Alignment | 26 | 18 | 31 | 27 | 20 | 16 |

| Instruction → Observation | Image → Image | Image → Video | Image → Point Cloud | Video → Image | Video → Video | Video → Point Cloud |
|---|---|---|---|---|---|---|
| **Any2Policy** | **56** | **63** | **38** | **46** | **57** | **36** |
| - Embodied Alignment | 15 | 29 | 28 | 15 | 9 | 11 |

of our approach across a wide range of instruction and observation modalities. Subsequently, we assess the performance of our method on three simulated benchmarks. Each benchmark is limited to one or a small number of modalities. This phase aims to establish that, by leveraging training on cross-modal information, our method can still deliver competitive or superior results compared to standard benchmarks, even in the presence of a single modality.

## 4.1 Real-World Evaluation

**Evaluation Setup.** We conduct our evaluations using a newly constructed dataset of multimodal instruction and observation. The dataset is divided into training, validation, and testing subsets, with a split of 7/1/2, respectively. We position the objects randomly, allowing tasks to be performed on unseen objects.

**Implementation details.** We use an initial learning rate of 3e-5 with the AdamW [107] optimizer, a weight decay of 1e-6, and a linearly decaying learning rate scheduler with a warm-up covering the initial 2% of the total training time [108]. We apply a gradient clipping of 1.0. All experiments are evaluated over 10 trials to obtain the mean success rate. The action space is the absolute joint position.

**Experiment Results.** We aim to answer the following questions with our experiments.

*1. Does Any2Policy outperform single-modality trained methods?* We compare Any2Policy with modality-specific approaches in Table 1. In particular, all implementations and network architecture remain constant across both settings, except that Any2Policy is trained on a variety of modalities while the modality-specific method is limited to particular ones. We observe that Any2Policy consistently achieves superior performance across all instruction-observation pairs compared to the modality-specific approach. Given that the total volume of training data is identical due to fixed training steps, this enhanced performance is attributed to learning from multiple modalities, which aids in better model generalization.

*2. The significance of embodied alignment.* The impact of embodied alignment is specifically explored in our experiments presented in Table 2. We experiment by removing the embodied alignment module and simply concatenating the outputs of projection layers from different modalities. TokenLearner is still utilized to manage the memory burden of numerous tokens. The alignment between instruction tokens and observation tokens is then achieved using an MLP layer, followed by the policy network. Notably, there is a significant performance drop when embodied alignment is omitted.

*3. Does Incorporating More Modalities Enhance Performance?* A key advantage of the Any2Policy framework is its ability to outperform modality-specific models by training on multiple modalities for instruction-observation pairs. It raises the question: can including more modalities at the inference stage further improve performance? Table 4 presents our experimental findings. We used the text-image pair as the baseline, a common setup in robotics. Adding additional instructional

Table 3: Comparison with state-of-the-art robotic models in real-world experiments.

| Instruction → Observation | Image → Image | Text → Image | Video → Image | Image + Text → Image |
|---|---|---|---|---|
| VIMA [59] | - | - | - | 49 |
| R3M [109] | 42 | - | 46 | - |
| T5 [110] | - | 39 | - | - |
| **Any2Policy** | **44**$_{+2}$ | **51**$_{+12}$ | **59**$_{+13}$ | **62**$_{+13}$ |

Table 4: Ablation study on the effect of using different modalities in real-world experiments.

| Method | Instruction | Observation | Success Rate |
|---|---|---|---|
| Any2Policy | Text | Image | 51 |
| | Text+Audio | Image | 52 |
| | Text+Audio+Image End-Goal | Image | 60 |
| | Text+Audio+Image End-Goal + Video Demonstration | Image | **62** |
| Any2Policy | Text | Image | 51 |
| | Text | Image+Video | 57 |
| | Text | Image+Video+Point Cloud | **66** |
| Any2Policy | Text | Image | 51 |
| | Text+Video Demonstration | Image+Video | 63 |
| | Text+Image End-Goal | Image+Point Cloud | **69** |

modalities, such as audio, image-based end-goals, and video demonstrations, consistently increased the success rate. Interestingly, although text and audio convey similar instructions, their combination still yielded a slight performance improvement of 1%. For observations, significant performance gains were observed when adding video and point cloud data, integrating temporal and 3D spatial information. Combining multiple instructional and observational modalities also improved performance; for example, incorporating video demonstrations in instructions and video in observations led to a 12% increase in the success rate.

*4. Comparison of Any2Policy with state-of-the-art robot models.* To further assess the effectiveness of Any2Policy representations, we compare them with other robot models, as shown in Table 3. Any2Policy consistently outperforms both T5 [110] and R3M [109] across various modalities, highlighting the advantage of incorporating multiple modalities during training. Furthermore, Any2Policy significantly surpasses VIMA [59], a recent method that uses text goals and object images for task specification. This demonstrates that Any2Policy not only outperforms single-modal counterparts, but is also highly effective in employing multiple modalities during inference.

*5. Other results.* We observe that the performance of image-point cloud and video-point cloud pairings achieves relatively lower scores compared to other modality pairs. However, when we use text or audio as instruction, the performance of point cloud is typically better than image or video due to its accurate position in three-dimensional space. This indicates that there is still room to improve model performance with point cloud inputs. These results highlight the inherent complexity of integrating visual modalities like images and videos with point cloud data, which often requires sophisticated fusion techniques to effectively combine spatial information with traditional 2D features.

## 4.2 Simulation Evaluation

**Dataset.** Our experiments are based on the following three datasets in our simulation. These three datasets represent three different instruction-observations modalities. Specifically, Franka Kitchen [92] uses text-image and ManiSkill2 [94] uses text-image and text-{image, point cloud}.

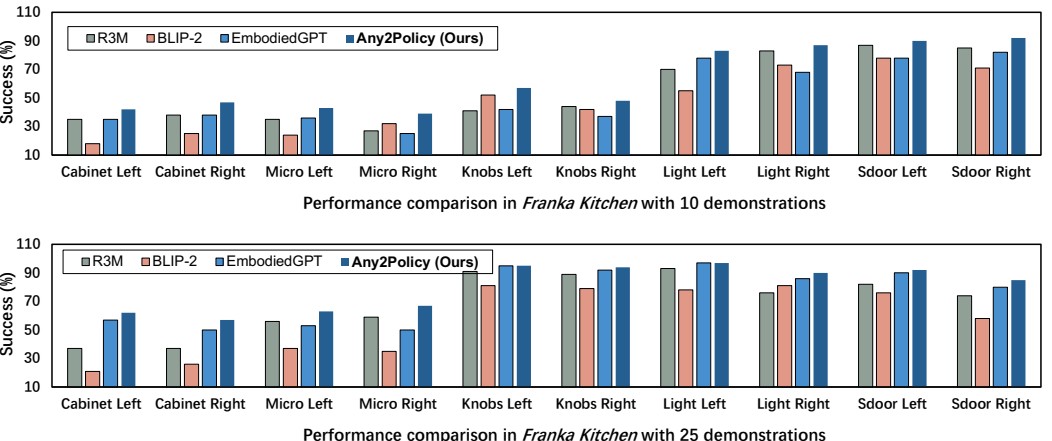

Figure 4: Performance of Any2Policy in Franka Kitchen with 10 or 25 demonstration demos. Comparison with R3M, BLIP-2, and EmbodiedGPT. On all tasks except for Knobs-left with 25 demonstrations, we obtained superior performance over SOTA methods.

The objective of our simulated environments is to evaluate the efficacy of our proposed method under conditions where modalities are limited.

**Implementation Details.** All models are trained on A100 GPUs, implemented in PyTorch [111]. The Franka-Kitchen are trained for 40K steps. We use weight decay of 1e-6, cosine learning rate scheduler with warmup steps of 2% total steps. The gradient clip of 1.0 is also applied. We use Adam optimizer with initial learning of 1e-3 and 3e-4 for Franka Kitchen and Maniskill-2. Note that Maniskill-2 does not have text instructions. We augment the task description into instruction via GPT-4. It is then tokenized and prepend with image backbone via FiLM [112].

**Evaluation**: We evaluate our approach using 30 rollouts from the BC learned policy. We use the mean success rate of the final policy as our metric. When providing task suite metrics we average the mean success rate across camera configurations.

**Experimental results on Franka Kitchen.** In our experiments, we benchmarked our model against two state-of-the-art methods: R3M [109], and BLIP-2 [22], a sota vision-language model, and Embodied-GPT [113], a multi-modal model designed for robotics. Our policy network was trained using few-shot learning, utilizing either ten or twenty-five demonstrations. We assessed the success rate of these models in 100 randomized evaluations across five different tasks in each benchmark. These evaluations were conducted under two settings, each with five separate runs and from two different camera perspectives, using only visual observations. The results, illustrated in Figures 4 for the Franka Kitchen, clearly show that Any2Policy outperforms the baseline methods, on both 10 and 25 demonstrations, except for Knobs-left with 25 demonstrations.

# 5  Conclusion

Humans interact with and perceive the world through information from multiple modalities. Existing research in robotics predominantly concentrates on policy learning using a single modality. In contrast, this paper introduces the Any2Policy framework, which adeptly demonstrates the integration of various sensory modalities into a cohesive model. This framework efficiently processes and responds to multi-modal data for robotic tasks. The overall framework, coupled with its multi-modal dataset, represents a significant advancement in the field of embodied AI. Our findings underscore the potential of Any2Policy to augment robotic adaptability and performance, thereby highlighting the importance of multi-modal methodologies in the development of more generalized and versatile robotic systems.

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
