# OpenReview forum: "Any2Policy: Learning Visuomotor Policy with Any-Modality"
_NeurIPS.cc/2024/Conference — NeurIPS 2024 poster_

### Official Review · Reviewer_5TLy · 2024-06-13

**Soundness:** 3
**Presentation:** 3
**Contribution:** 3
**Rating:** 6
**Confidence:** 4

**Summary:**

This paper aims to enable robots to understand tasks and their environments using multiple modalities such as text, audio, images, video, and point clouds. To accomplish this, the authors introduce Any2Policy, a versatile framework designed to process multi-modal inputs at the instruction and observation levels. Within this framework, they developed an embodied alignment module that effectively integrates multi-modal features. Additionally, the use of ImageBind and TokenLearner helps to mitigate tedious encoder design and enhance training efficiency, respectively, which appears to be a practical approach. The paper's thorough evaluations, conducted on 30 real-world tasks and three distinct simulation benchmarks, convincingly demonstrate the effectiveness of the proposed method.

**Strengths:**

1. This paper addresses a pivotal topic in robot learning, focusing on enabling robots to understand tasks and interpret the environment through multi-modal inputs, which is essential for developing generalist agents.

2. The manuscript is well-written and well-motivated, presenting its ideas and research goals effectively.

3. The methods introduced in this paper effectively integrate multi-modal inputs to substantially improve policy learning. Comprehensive evaluations conducted across both real-world tasks and simulated environments demonstrate the effectiveness of the Any2Policy framework.

**Weaknesses:**

However, there are still some concerns regarding the experiments.

1. While the authors have conducted extensive ablations on real-world experiments, replicating these results in practice may be challenging for follow-up researchers. To facilitate better benchmarking, it would be advantageous if the effectiveness of each proposed method could also be demonstrated in simulated experiments.

2. Due to the high cost associated with evaluating real-world tasks across different random seeds, reporting variance is difficult. Nonetheless, it would be beneficial if the authors could assess the impact of multiple random seeds in simulation tasks to provide more robust statistical insights.

3. The authors appear to have omitted the results from the meta-world evaluations. These results were neither found in the main text of the paper nor in the appendix, despite the authors claims that they were reported.

4. MUTEX [1] is another significant work that explores task specifications using multi-modal inputs. While Any2Policy incorporates a broader range of observation modalities (image, point cloud), including the comparison with MUTEX could enrich the analysis, especially given the limited range of baseline methods evaluated in this paper.


[1]  Mutex: Learning unified policies from multimodal task specifications

**Questions:**

1. I'm curious to know whether the video instructions were recorded using the same embodiment or if they include cross-embodiment examples, such as demonstrations by humans?

**Limitations:**

The authors have properly discussed the limiations and societal impacts in the Appendix.

---

> ### Author Rebuttal · Authors · 2024-08-07
>
> We thank reviewer for the comments and reference reviewer with identifier 5TLy as R4. Comment n of reviewr m is denoted as RmCn.
>
> **[R4C1]**
> > While the authors have conducted extensive ablations on real-world experiments, replicating these results in practice may be challenging for follow-up researchers. To facilitate better benchmarking, it would be advantageous if the effectiveness of each proposed method could also be demonstrated in simulated experiments.
>
> We share the reviewer's sentiment that building a simulation environment is a paramount topic that could improve productivity and enhance future research. We are currently developing a simulation based on Isaac Sim, which we hope will facilitate further research in this domain.
>
> **[R4C2]**
>
> > Due to the high cost associated with evaluating real-world tasks across different random seeds, reporting variance is difficult. Nonetheless, it would be beneficial if the authors could assess the impact of multiple random seeds in simulation tasks to **provide more robust statistical insights**.
>
> Since our simulation environments were not ready, we performed a robustness study on random seeds in real-world experiments. We evaluated four tasks in the real world, running 3 seeds for each experiment with seed numbers 0, 1, and 2. For each seed, we conducted 10 evaluations across the four real-world tasks. Both methods were evaluated 10 times, and we report the average success rate for each method.
>
> | Seed | PlaceBread | CloseLaptop | InsertPlug | PlaceCan |
> | ------- | ------- | ------- | ------- | ------- |
> | 1 | 100 | 100| 50 | 50 |
> | 2 | 90 | 100| 30 | 50 |
> | 3 | 100 | 90| 50 | 60 |
> | Average| 96.6 | 96.6 | 43.3 | 53.3 |
>
> Our experiments indicate that the training of our approach is stable, and the results are consistent across different seeds.
>
> **[R4C3]**
>
> > Missing MetaWorld Results.
>
> We thank Reviewer 4 for the careful review and for pointing out the missing experimental results. We sincerely apologize for this oversight. Here is a brief summary of the experimental results. We present the results based on three task levels, following the settings in Masked World Model (CoRL'23). We report the average success rate across three task levels: easy, medium, and hard. In total, there are 45 tasks. All experiments were trained with 20 demonstrations, evaluated with 3 seeds, and for each seed, the success rate was averaged over five different iterations. We used all available modalities for our method. The experimental results are shown below:
>
> | Method | Easy (28) | Medium (11) | Hard (6) |
> | ------- | ------- | ------- | ------- |
> | Diffusion Policy (RSS'23) | 80.4 | 32.6 | 9.4 |
> | IBC (CoRL'21) | 68.3 | 15.2 | 10.6
> | EmbodiedGPT (NeurIPS'23) | 60.1 | 24.5 | 7.9 |
> | **Ours** | **89.6** | **65.3**| **29.6** |
>
> The numbers in parentheses indicate the number of tasks for each specific task level. The experimental results support our conclusion that Any2Policy are able to achieve better performance than the baseline across different levels of task difficulty.
>
> **[R4C4]**
>
> > Comparison with Mutex.
>
> We thank the reviewer for raising this point. Mutex is indeed a significant work that explores multimodal task specification. To strengthen our work, we conducted a comparative study with Mutex. Specifically, we selected four real-robot tasks and compared them with Mutex using the same training data. Both methods were evaluated 10 times, and we report the average success rate of each method.
>
> | Method | PlaceBread | CloseLaptop | InsertPlug | PlaceCan |
> | ------- | ------- | ------- | ------- | ------- |
> | Mutex |   80 | 90 |  20 | 10 |
> | **Ours** | **100** | **100**| **50** | **50** |
>
> Our experimental results indicate that Any2Policy performs better than Mutex. We believe that the performance gain comes from the additional modalities introduced by our method.
>
>
> **[R4C5]**
> > I'm curious to know whether the video instructions were recorded using the same embodiment or if they include cross-embodiment examples, such as demonstrations by humans?
>
> Yes, the video instructions were recorded through human demonstration.

---

> ### Comment · Reviewer_5TLy · 2024-08-08
>
> Thanks for the careful comments. My major concerns are well-resolved. So, I'm happy to increase my confidence to reflect this (from 3 to 4).

---

### Official Review · Reviewer_RRf1 · 2024-06-18

**Soundness:** 3
**Presentation:** 3
**Contribution:** 2
**Rating:** 5
**Confidence:** 4

**Summary:**

This paper proposes the simultaneous fusion of image, text, point cloud, video, and audio—four modalities—in robotic manipulation tasks, while also considering the integration of information from both instruction and observation. Through the transformer architecture and cross-attention mechanism, embodied alignment of multiple modalities is achieved. Furthermore, this paper constructs a real multimodal dataset and conducts a detailed experimental analysis of the impact of additional modality information on performance.

**Strengths:**

1. This paper investigates the impact of simultaneously fusing five modalities: image, text, point cloud, and video, on robotic manipulation tasks. Although there is already a considerable amount of work that fuses two or three of these modalities, this paper argues that it is still meaningful to consider the simultaneous fusion of different modalities from both instruction and observation.

2. This paper constructs a real robotic dataset that includes information from all the aforementioned modalities.

3. The experimental section of this paper provides a detailed analysis of the impact of multimodal information fusion on robotic manipulation, confirming that the involvement of additional modalities in training or inference can lead to performance improvements.

**Weaknesses:**

1. The technical contribution of this paper is limited. Although the paper analyzes some difficulties in the process of fusing multimodal information and embodied alignment, the overall model architecture is still relatively straightforward and simple.

2. The scale of the dataset is relatively small compared to the widely used robotic operation datasets currently available. Considering that the model architecture and training methods for multimodal information fusion typically require a large amount of data, it may be necessary to have a larger dataset to verify stronger generalizability or to draw more robust conclusions on multimodal information fusion. Additionally, some existing datasets are already capable of including modalities such as text, image, video, and point cloud simultaneously. The main contribution of this paper seems to lie in the addition of rich paired audio data. If one considers expanding the scale of the dataset as well as costs, would it be a better choice to augment the existing datasets with information from modalities such as audio?

3. The experimental section does not yield particularly new conclusions. Generally speaking, it is not surprising that the addition of new modalities, especially for robotic manipulation tasks, typically leads to better performance.

**Questions:**

Considering the excellent development of current multimodal pre-trained models, and that the models in this paper are trained from scratch, is it feasible to incorporate existing pre-trained models such as text-image, audio-image, image-point cloud, and text-point cloud models into the current training framework?

**Limitations:**

The authors have adequately addressed the limitations andsocietal impact of the work.

---

> ### Author Rebuttal · Authors · 2024-08-07
>
> Rebuttal:
>
>
> We thank reviewer for the comments and reference reviewer with identifier RRf1 as R3. Comment n of reviewr m is denoted as RmCn.
>
> **[R3C1]**
> > The technical contribution of this paper is limited. Although the paper analyzes some difficulties in the process of fusing multimodal information and embodied alignment, the overall model architecture is still relatively straightforward and simple.
>
> We thank the reviewer for raising this point. While the technical novelty may seem limited, our primary contribution lies in providing a straightforward yet effective solution that, for the first time, successfully integrates multiple modalities for both task specification and observation, enabling seamless robotic manipulation. Our method demonstrates significant performance gains over the baselines in new settings, underscoring the necessity of our approach
>
>
>
> **[R3C2]**
> > The proposed dataset seems to lie in the addition of rich paired audio data? Would it be a better choice to augment the existing datasets with information from modalities such as audio?
>
> We appreciate the reviewer's concern about our dataset. Notably, our dataset goes beyond merely adding rich paired audio data. We offer a comparison with existing datasets in the following table. We focus on the types of modalities in existing data on both observation and task specification.
>
> | Dataset | Observation | Task Specification |
> | ------- | -----------  | ------------------|
> | Open-X Embodiment | point cloud, image | text |
> | Droid | point cloud, image | text |
> | RH20T | point cloud, image | video, text |
> | **Ours** | point cloud, image, video | text, image, video, audio |
>
> Most previous datasets provide point clouds and images for observation, and text for task specification. There are other modalities, such as image goals and audio, for task specification. Despite being smaller than Open-X and other datasets, our dataset is more comprehensive in the domain we aim to study, specifically aligning multiple modalities for both instruction and observation to facilitate the training of robotic manipulation. It highlights the unique value our dataset brings to the community.
>
> Secondly, while it is possible to expand existing datasets with this additional information, it would require significant effort and time. This presents a valuable future research direction for expanding datasets to include richer multimodal information.
>
>
>
> **[R3C3]**
>
> > The experimental section does not yield particularly new conclusions. Generally speaking, it is not surprising that the addition of new modalities, especially for robotic manipulation tasks, typically leads to better performance.
>
> We agree with the reviewer's point. Intuitively, adding new modalities should improve model performance for manipulation tasks. However, prior works have not fully explored this area. The main contribution of our work lies in building a robust system for multimodal settings, providing a benchmark to evaluate methods in this context, and demonstrating the value and potential of this approach for future research.
>
>
>
> **[R3C4]**
> > Considering the excellent development of current multimodal pre-trained models, and that the models in this paper are trained from scratch, is it feasible to incorporate existing pre-trained models such as text-image, audio-image, image-point cloud, and text-point cloud models into the current training framework?
>
> The challenge here is that an embodied agent needs to _observe_ and _interact_ with the world using different modalities. Not only do we need to manage diverse modalities for tasks such as observation and task specification, but we also need to align these two components together. Currently, no work addresses this comprehensive challenge, and it is infeasible to incorporate existing pre-trained models to resolve it.

---

> > ### Comment · Reviewer_RRf1 · 2024-08-11
> >
> > Thanks for the authors' sincere response. Despite the objective shortcomings that I mentioned, I generally recognize the contribution and solidity of this paper. Therefore, I maintain a positive attitude towards this paper overall, and I will discuss the review results with other reviewers in the follow-up.

---

### Official Review · Reviewer_QY6x · 2024-07-12

**Soundness:** 3
**Presentation:** 3
**Contribution:** 2
**Rating:** 6
**Confidence:** 5

**Summary:**

This submission develops a new model that can handle many different modalities as input for instruction and observations (video, text, image, pointcloud etc.). Different modalities are encoded via different frozen encoders (with projection layers kept trainable) to a shared representation to then be passed into a policy that then generates actions. The submission constructs a multi-modal dataset for several evaluation benchmarks (3 simulated, 1 real), and trains a policy on this multi-modal data and show it out-performs policies trained on a specific modality of data. While work in the past has built multi-modal models in language/vision, this is the first to do so in robotics. Cross attention is used where the observation sequence of tokens are keys and the instruction sequence of tokens are values.

**Strengths:**

- This appears to be the first work that combines video, image, pointcloud, and audio together into one multi-task model for robotics, demonstrating some improvements in success rates when multiple modalities are used in training instead of just one modality.
- The release of a highly multi-modal annotated dataset is great to see. However more details about the data would be greatly appreciated.

**Weaknesses:**

- The performance of AnyPolicy looks very similar to EmbodiedGPT in the franka kitchen setting in Figure 4. The figure 4 caption appears to also be wrong as it seems EmbodiedGPT outperforms AnyPolicy on both Knobs Left and Light Left, not just Knobs left. Moreover, no error bars are shown and with results so similar over just 100 evaluations in simulation, it is hard to say if AnyPolicy out performs embodied GPT.
- FrankaKitchen, while a useable benchmark, has almost no initial randomization and is incredibly easy given objects in the environment are not randomized, making results reported on it not mean that much. It would be great to see comparisons on harder manipulation benchmarks although I realize it might not be possible to run experiments now.
- Very few details about the real world dataset are provided. What are the 30 tasks? What are some example instructions? What are some example audio?

**Questions:**

Questions:
- Are there meta-world results? I can't find any figures/tables on it, only see results on ManiSkill2 and Franka Kitchen.
- The name of the dataset this paper curated, RoboAny, is mentioned once in the paper in passing and never again. Is there a reason for that?
- Can there be more clarification around how the ablation of no embodied alignment module works? My understanding is if you remove it you simply directly use the encoded tokens and put them all in one big sequence as input to the transformer. However the original encoders might output different shapes for the tokens so I am not sure how this is consolidated.
- Is AnyPolicy trained on data from all benchmarks or is a separate policy trained per benchmark. In other words, is the multi-task policy multi-task over all tasks in a benchmark, or all tasks in all benchmarks?
- How does the modality-specific model work, what is its architecture compared to AnyPolicy?
- How come StackCube has such a high success rate but PickCube has such a low success rate? My understanding is in ManiSkill2 StackCube is much harder than PickCube because it requires careful placement on a cube and releasing the cube, whereas in PickCube you simply grab the cube and move to a random goal.
- How many demonstrations are used for ManiSkill2? This detail could not be found anywhere.
- How is R3M used in the Video-Image result in table 3? How does it process video input and produce video tokens?

Typo:
- "We further conduct experiments on two simulated robotics benchmarks to reinforce the strong generalizability of our approach compared to existing methods" in section 1. I think three benchmarks were tested not two.
- Figure 4 shows 110% on the y-axis. It should be clamped to [0, 100].

Unfortunately given the significant lack of details on data, and how some ablations work (see questions below for more) I recommend a reject in the current state. Moreover the performance of the model after combining several new modalities does not appear to out perform other baselines from other papers that much to warrant the amount of extra work on a simple task like FrankaKitchen. I am happy to raise my score to the accept range if these weaknesses and questions are addressed.

**Limitations:**

Limitations are provided in the appendix and discuss limitations from the perspective of additional things that could be done (more modalities, more robots etc.). I think limitations around data efficiency could be important to bring up. Just how many demonstrations are needed? Real-world imitation learning suffers greatly due to the expensive cost of teleoperation (especially if high success rates are desired).

---

> ### Author Rebuttal · Authors · 2024-08-07
>
> We thank reviewer for the comments and reference reviewer with identifier QY6x as R2. Comment n of reviewr m is denoted as QY6x.
>
> **[R2C1]**
>
> > The performance of AnyPolicy looks very similar to EmbodiedGPT in the Franka kitchen setting in Figure 4. The Figure 4 caption appears to be wrong
>
> Thank you for pointing out our mistake. The caption is indeed incorrect, as Any2Policy performs slightly worse than (or on par with) EmbodiedGPT on two tasks. However, we respectfully disagree that the overall performance of Any2Policy is similar to EmbodiedGPT. For the Franka Kitchen experiments, we demonstrate that:
>
> 1. For challenging tasks, such as Micro Right and Cabinet Right, our method significantly outperforms EmbodiedGPT.
> 2. In low-data scenarios (e.g., 10 demonstrations), Any2Policy achieves much better performance than EmbodiedGPT and other baselines.
> 3. Despite being developed to handle multiple modalities, our method outperforms those trained specifically for a single modality.
>
>
> **[R2C2]**
>
> > The experiments of FrankaKitchen is an easy benchmark.
>
> We agree that FrankaKitchen is an easy benchmark and may introduce bias in evaluation. Therefore, we have provided further evaluation on Meta-World (results are included in the rebuttal), ManiSkill-2, and real robots. To further address the reviewers' concerns, we conducted additional experiments on RLBench and Adroit, which include more challenging tasks with a 24 DoF ShadowHand and a 4 DoF arm. All experiments were conducted with 200 demonstrations.
>
> We provide multiple baselines, including Diffusion Policy, IBC, EmbodiedGPT, and also present error bars. We ran each experiment with 3 seeds and evaluated 200 episodes every 200 training epochs. We computed the average of the highest 5 success rates, which is a typical setting in this simulation. Below are the experimental results.
>
>
> | Method | Hammer | Door | Pen |
> | ------- | ------- | ------- | ------- |
> | Diffusion Policy | 52 $\pm$ 15| 56 $\pm$ 7 |  17 $\pm$ 5 |
> | IBC | 0 $\pm$ 0 | 1 $\pm$ 1 | 0 $\pm$ 0 |
> | EmbodiedGPT | 35 $\pm$ 4 | 49 $\pm$ 10 |  21 $\pm$ 7 |
> | **Ours**|  **64 $\pm$ 13** | **65 $\pm$ 8** | **29 $\pm$ 6** |
>
>
> We demonstrate that even on challenging benchmarks, our method consistently outperforms state-of-the-art baselines.
>
>
> **[R2C3]**
>
> > Metaworld Results are missing.
>
> We thank Reviewer 4 for the careful review and for pointing out the missing experimental results. We sincerely apologize for this oversight. Due to limited space, please see **[R4C3]** for details.
>
>
> **[R2C4]**
>
> > Name Roboany is not used anywhere else.
>
> We thank R4 for careful reviews and for pointing out this **typo**. We will include this in the revision.
>
> **[R2C5]**
>
> > Can there be more clarification around how the ablation of no embodied alignment module works?
>
> We appreciate the reviewer's concern regarding the details of our ablation study. We have provided a **detailed illustration of our methodology in lines 254-258**. Specifically, we use MLP layers to align the observation tokens with the instruction tokens, ensuring the dimensions are correct. Additionally, we employ TokenLearner to reduce the number of tokens and maintain consistency, even when some modalities are missing.
>
>
> **[R2C6]**
>
> > Is AnyPolicy trained on data from all benchmarks or is a separate policy trained per benchmark. In other words, is the multi-task policy multi-task over all tasks in a benchmark, or all tasks in all benchmarks?
>
> In our experiments, for each benchmark, including real-world scenarios, our model is trained on all collected tasks to perform multi-tasking. This is a typical setting in robot learning, as seen in works like Diffusion Policy (RSS '23) and EmbodiedGPT (NeurIPS '23).
>
>
> **[R2C7]**
> > How does the modality-specific model work, what is its architecture compared to AnyPolicy?
>
> Due to our systematic design, our method can naturally handle missing modalities. The architecture of our modality-specific model is identical to AnyPolicy. During inference, we pass empty tokens to account for the missing modalities.
>
>
> **[R2C8]**
> > Since **PickCube only needs to place a cube on a random place**, which is much easier than StackCube, why does StackCube have such a higher success rate than PickCube?
>
> We thank Reviewer 4 for their careful review and for pointing out this observation. To explain, first of all, PickCube requires picking up a cube and placing it at a **specific goal position**. This task is not easier than StackCube. For instance, as shown in Table 3 of the ManiSkill-2 paper, **the success rate for StackCube is higher than for PickCube when using point cloud observations**. Secondly, since our method is trained in a multi-task fashion (as opposed to the single-task training in ManiSkill-2), the learning behavior could differ from that of a single-task policy.
>
> **[R2C9]**
> > How many demonstrations are used for Maniskill2?
>
> We used 1K demonstrations for all tasks for Maniskill2.
>
> **[R2C10]**
> > How is R3M used in the Video-Image results?
>
> For R3M, we select the keyframe with images of resolution $224 \times 224$ as input and pass to R3M. All other settings are kept the same as the default R3M.
>
> We thank you for the precious review time and valuable comments. We have provided corresponding responses, which we believe have covered your concerns. We hope to further discuss with you whether or not your concerns have been addressed. Please let us know if you still have any unclear parts of our work.

---

> > ### Comment · Reviewer_QY6x · 2024-08-10
> > **Response**
> >
> > Thanks for the comprehensive response.
> >
> > R2C2: I'm aware you test on other benchmarks, but surprised you keep FrankaKitchen in the main text and then the harder ManiSkill 2 benchmark in the supplemental (many people often miss the supplemental). Is there a reason for this? At minimum since there are MetaWorld results you can include those in the main text (they are far more impressive and significant, larger differences in success rates on harder tasks with a bit more diversity than franka kitchen).
> >
> > R2C10: What do you mean keyframe, how are keyframes selected? If these are hand picked or by some algorithm, could it be that the keyframe selection is poor leading to worse results?
> >
> > All my concerns are essentially covered. My main concern left really is the paper does not seem to be polished given some key figures were missing to begin with (and are only present in openreview comments, not revised pdfs submitted here).
> >
> > I have raised my score to 6 to reflect that I do wish to accept the paper. It is straightforward and fairly simple, easy to understand, and even if the results are highly expected, someone had to do the experiments and I really appreciate the efforts, I imagine a good amount of engineering is necessary to get this kind of project working. But I think its presentation and organization could be better. A big presentation issue for example is with regards to the choice to only include franka kitchen in the main text, without including MetaWorld or ManiSkill2.

---

### Official Review · Reviewer_h8z5 · 2024-07-22

**Soundness:** 3
**Presentation:** 3
**Contribution:** 2
**Rating:** 6
**Confidence:** 3

**Summary:**

The paper aims to enhance the generalizability of robotic agents by enabling them to handle tasks using diverse modalities such as text, audio, images, and point clouds. The authors introduce a multi-modal system named Any-to-Policy, which utilizes a versatile modality network to adapt various inputs and connects with policy networks for effective control. To support this approach, they assembled a comprehensive dataset covering 30 robotic tasks annotated across multiple modalities. Extensive experiments demonstrate the system's capability to adapt to diverse multi-modal inputs in both simulated and real-world settings, showcasing its potential to improve the generalizability and performance of embodied agents in various scenarios when comparing various baselines.

**Strengths:**

The paper introduces a novel approach to multi-modal robotic learning with the Any2Policy framework. This framework allows for the integration and processing of diverse modalities such as text, audio, images, and point clouds in a unified manner. In my opinion, the originality seems to lie in the seamless combination of these modalities, enabling robots to handle a wide variety of tasks with greater adaptability. The assembly of a comprehensive real-world dataset annotated across multiple modalities is also a significant and innovative contribution, addressing the scarcity of such datasets in the field.

Moreover, the authors provide detailed descriptions of their framework, including the use of multi-modal encoders and embodied alignment techniques. The experimental setup is relatively comprehensive, encompassing both simulated benchmarks and real-world scenarios, which validates the effectiveness of the proposed approach.

Finally, the paper addresses an important challenge in robotic learning: the ability to process and integrate information from multiple modalities. By demonstrating the capability of the Any2Policy framework to generalize across various tasks and modalities, the paper opens up new possibilities for the development of more generalizable and adaptive robotic systems. The release of the multi-modal dataset will likely push further research in this area, which is important for future work along this direction

**Weaknesses:**

1. While the assembly of a multi-modal dataset is a significant contribution, the paper could improve by providing more detailed information on the diversity and representativeness of the tasks and scenarios included in the dataset. For instance, it would be beneficial to know more about the variation in objects, environments, and task complexities. This information would help to assess whether the dataset sufficiently covers the wide range of scenarios the framework is intended to handle.

2. Although the paper compares the Any2Policy framework with several state-of-the-art models, it could benefit from a more extensive comparative analysis. For instance, including a wider variety of baseline models e.g. RT-1, RT-2, RT-X, Octo, OpenVLA etc. and providing a detailed discussion on the differences in performance could help to better highlight the strengths and weaknesses of the proposed approach. Additionally, a qualitative comparison, showing example outputs or behaviors from different models, could provide valuable insights.

3. The paper includes some ablation studies to assess the impact of different components of the framework. However, these studies could be expanded to provide a deeper understanding of the contribution of each component given that Any2Policy is a fairly complex system. For example, the authors could investigate the impact of different encoder architectures or the role of specific modalities in greater detail. This would help to identify the most critical elements of the framework and guide future improvements.

**Questions:**

See the section above.

**Limitations:**

Yes

---

> ### Author Rebuttal · Authors · 2024-08-07
>
> We thank reviewer for the comments and reference reviewer with identifier h8z5 as R1. Comment n of reviewr m is denoted as RmCn.
>
> **[R1C1]**
>
> > More details on the datasets.
> We concur with the reviewer's suggestion to provide additional details in our paper. To address this, we have included in the Appendix examples of 8 tasks from our real-world dataset. This dataset comprises 20 short-horizon tasks, such as pick-and-place [item], and 10 medium-to-long-horizon tasks, which involve multiple steps to complete (e.g., opening a drawer, retrieving an item, placing it back, and closing the drawer). Our setup features a variety of real-world objects, including microwaves, drawers, laptops, boxes, cubes, bottles, etc. These objects range from deformable to rigid and articulated, encompassing a wide array of real-world scenarios. We will provide a detailed summary of our dataset in the revised manuscript.
>
> **[R1C2]**
>
> > Extensive comparative analysis would be beneficial, such as RT-1/RT-2/RT-X, Octo, OpenVLA.
>
>
> We appreciate the reviewer for highlighting these issues. Since RT-2/RT-X are not open-sourced, and OpenVLA is released recently which we do not have time to implement in our environment, we provide additional results of Octo and RT-1 in our real world experiments. Given the limited time available for the rebuttal, we conducted experiments on four selected tasks. The same training data were used for all methods. For Octo, we present two versions: one that uses pretrained weights from the Open-X framework, and another that is trained from scratch using our dataset. Both versions were evaluated 10 times to calculate the average success rate for each method. Consistent training hyperparameters, including learning rate, learning rate scheduler, and training epochs, were maintained across all methods.
>
>
> | Method | PlaceBread | CloseLaptop | InsertPlug | PlaceCan |
> | ------- | ------- | ------- | ------- | ------- |
> | Octo (pretrained) |   **100** | **100** |  40 | 20 |
> | Octo (train-from-scratch) |  50 | 70 |  0 | 0 |
> | RT-1  |  60 | 80 |  0 | 0 |
> | **Ours** | **100** | **100**| **50** | **50** |
>
>
> We demonstrate that Octo, pre-trained on the Open-X framework, achieves comparable results to our method on two tasks. However, for more challenging tasks, our method outperforms the pre-trained Octo. When Octo is trained with the same amount of data as our method, our approach achieves a significantly higher success rate.
>
> **[R1C3]**
>
> > The author could investigate the impact of different encoder architectures or the role of specific modalities in greater detail.
>
> We thank the reviewer for raising these points. Unfortunately, due to the limited time available during the rebuttal period, we were unable to complete these experiments. However, we plan to conduct additional experiments in the future, focusing on using different encoders and replacing the policy head.

---

### Decision · Program_Chairs · 2024-09-25

**Decision:**

Accept (poster)

**Comment:**

This paper presents a multi-modal robotic manipulation dataset encompassing five input modalities: image, text, audio, point cloud, video. This paper demonstrates that a simple approach integrating all these modalities lead to improved performance in robotic manipulation tasks.

Despite its limited technical novelty, the reviewers have reached a consensus that the curated dataset and comprehensive empirical evaluations offer valuable insights to the robotic learning community. As such, this paper is recommended for acceptance to NeurIPS.

However, most of the reviewers have concerns on this paper's presentation and organization. The author responses during the rebuttal period have resolved some of these issues, but it does not seem sufficient. The authors should carefully address the reviewers' comments and improve the quality of the paper together with more comprehensive experimental results beyond the ones included in the rebuttal.